# The Role of Palmitoleic Acid in Regulating Hepatic Gluconeogenesis through SIRT3 in Obese Mice

**DOI:** 10.3390/nu14071482

**Published:** 2022-04-01

**Authors:** Xin Guo, Xiaofan Jiang, Keyun Chen, Qijian Liang, Shixiu Zhang, Juan Zheng, Xiaomin Ma, Hongmei Jiang, Hao Wu, Qiang Tong

**Affiliations:** 1Department of Nutrition and Food Hygiene, School of Public Health, Cheeloo College of Medicine, Shandong University, Jinan 250012, China; 201835803@mail.sdu.edu.cn (X.J.); lqj@mail.sdu.edu.cn (Q.L.); amyzhangsx@sdu.edu.cn (S.Z.); mxm@sdu.edu.cn (X.M.); jianghongmei@sdu.edu.cn (H.J.); hwu@sdu.edu.cn (H.W.); 2Department of Pediatrics, USDA/ARS Children’s Nutrition Research Center, Baylor College of Medicine, Houston, TX 77030, USA; heather.chen@yahoo.com; 3Department of Endocrinology, Union Hospital, Tongji Medical College, Huazhong University of Science and Technology, Wuhan 430022, China; zhengjuan25@163.com; 4Hubei Provincial Clinical Research Center for Diabetes and Metabolic Disorders, Wuhan 430022, China; 5Department of Molecular Physiology & Biophysics, Department of Medicine, Huffington Center on Aging, Houston, TX 77030, USA

**Keywords:** gluconeogenesis, high-fat diet, palmitoleic acid, SIRT3

## Abstract

Hepatic gluconeogenesis is a crucial process to maintain glucose level during starvation. However, unabated glucose production in diabetic patients is a major contributor to hyperglycemia. Palmitoleic acid is a monounsaturated fatty acid (16:1n7) that is available from dietary sources. Palmitoleic acid exhibits health beneficial effects on diabetes, insulin resistance, inflammation, and metabolic syndrome. However, the mechanism by which palmitoleate reduces blood glucose is still unclear. SIRT3 is a key metabolism-regulating NAD^+^-dependent protein deacetylase. It is known that fasting elevates the expression of SIRT3 in the liver and it regulates many aspects of liver’s response to nutrient deprivation, such as fatty acid oxidation and ketone body formation. However, it is unknown whether SIRT3 also regulates gluconeogenesis. Our study revealed that palmitoleic acid reduced hepatic gluconeogenesis and the expression of SIRT3 under high-fat diet conditions. Overexpression of SIRT3 in the liver and hepatocytes enhanced gluconeogenesis. Further study revealed that SIRT3 played a role in enhancing the activities of gluconeogenic enzymes, such as PEPCK, PC, and MDH2. Therefore, our study indicated that under a high-fat diet, palmitoleic acid decreased gluconeogenesis by reducing enzymatic activities of PEPCK, PC, and MDH2 by down-regulating the expression of SIRT3.

## 1. Introduction

Type 2 diabetes is currently a grave health threat to the world’s population. Approximately 1.47 million people die from diabetes and diabetic complications every year [1]. Diabetes patients are also at a high risk of cardiovascular disease [2]. The global medical expenditure for diabetes and related complications is a heavy burden to the economy. Nutritional intervention is an economical and effective way to prevent and manage diabetes. The liver is an important organ for metabolism. Hepatic gluconeogenesis is the process of converting various non-sugar substances into glucose. In the development of type 2 diabetes, elevated hepatic gluconeogenesis is very common. Although there is no difference in basic gluconeogenesis among diabetic patients, gluconeogenesis is not reduced after food intake [3]. The increase in hepatic gluconeogenesis is one of the important pathogenic factors in the development of diabetes. Therefore, exploring the regulatory mechanism of hepatic gluconeogenesis and finding nutritional interventions are of great significance in the prevention and management of type 2 diabetes.

Palmitoleic acid (PO) is a 16-carbon monounsaturated fatty acid (16:1n7) that can be obtained from the diet. It is mainly derived from marine products and is also present in breast milk, animal fats, and vegetable oils. Endogenous palmitoleic acid can also be produced by adipocytes [4]. Palmitoleic acid has a positive preventive effect on diabetes, insulin resistance, inflammation, and metabolic syndrome [5,6,7]. Palmitoleic acid is considered to be a biologically active lipid that regulates the metabolic link between the liver and fat. Administration of palmitoleic acid by gavage in mice can increase the level of palmitoleic acid in the blood circulation and increase systemic insulin sensitivity. Palmitoleic acid reduces the number of macrophages in the liver and reduces the inflammatory response by inhibiting the phosphorylation of nuclear factor κB p65. When primary hepatocytes and macrophages are treated with palmitoleic acid, the phosphorylation of nuclear factor kappa B p65 is inhibited, and the release of inflammatory factors, including tumor necrosis factor alpha and interleukin 6, are reduced. In addition, palmitoleic acid increases liver fat deposition but inhibits liver inflammation and increases liver insulin sensitivity [8]. In mouse liver and primary hepatocytes, palmitoleic acid can promote insulin action by increasing the phosphorylation of Akt (Ser473) [8]. Palmitoleic acid can also increase insulin signaling in muscles [7] and the intake and utilization of glucose in fat cells [9], thereby reducing blood glucose levels. However, whether palmitoleic acid can affect the synthesis of glucose in the liver has not been reported yet.

Sirtuins are NAD^+^-dependent protein deacetylases. Sirtuin 3 (SIRT3) is one of the seven members of sirtuins (SIRT1-7) in mammals [10]. SIRT3 is highly expressed in major metabolic organs, such as brown adipose tissue [11], muscle [12], heart [13], and liver [14]. SIRT3 mainly exists in mitochondria; however, some studies have shown that SIRT3 also binds and deacetylates proteins in the cytoplasm and nucleus [15]. During nutrient deprivation, SIRT3 exerts extensive metabolic regulation by deacetylating many mitochondrial enzymes. These mitochondrial enzymes include long-chain acyl-CoA dehydrogenase in fatty acid oxidation, HMG coenzyme A synthetase 2 in ketogenesis, acetyl-CoA synthase 2 in acetate consumption, and ornithine transcarbamylase in urea formation. SIRT3 also regulates enzymes in the respiratory chain in mitochondria. Therefore, SIRT3 closely regulates energy metabolism [16,17,18]. In addition, SIRT3 promotes the conversion of muscle fiber types [12,19] and lipid utilization [20,21]. In brown fat, SIRT3 increases adaptive thermogenesis by activating mitochondrial biosynthesis and respiration in response to food restriction and cold exposure [11].

In our previous study, we found that HFD-fed mice treated with palmitoleic acid had reduced HFD-induced liver insulin resistance and inflammation [8]. In this study, we found palmitoleic acid inhibited hepatic gluconeogenesis and increased systemic insulin sensitivity. Furthermore, we found palmitoleic acid suppressed the expression of SIRT3 and gluconeogenic enzyme activities in the liver.

## 2. Materials and Methods

### 2.1. Preparation of SIRT3 Expression

AAV-SIRT3M1-FLAG (AAV-SIRT3) cDNA was excised from pcDNA3.1-SIRT3M1-FLAG with XbaI, followed by Klenow treatment and then Kpn I digestion. The fragment was inserted into Hind II and Kpn I sites of AAV-TBG-eGFP vector to generate AAV-TBG-eGFP-SIRT3M1-FLAG. The construct was confirmed by sequencing. AAV-TBG-eGFP (AAV-GFP) was used as the control vector. AAV8 production and purification of the viruses were performed by OBiO Technology Company (Shanghai, China). Adenoviral vectors Ad-GFP and Ad-SIRT3 (with the GFP label) [22] were produced by OBio Technology Company (Shanghai, China).

### 2.2. Animal Experiments

Male C57BL/6N mice were maintained on a 12:12-h light-dark cycle (lights on at 06:00). Sixty mice aged 5–6 weeks purchased from Beijing Vital River Laboratory Animal Technology Corporation (Beijing, China) were adapted for one week, were randomly divided into 2 groups (30 mice per group), and were fed with a high-fat diet (HFD) (D12492) or low-fat diet (LFD) (MD12450B) for 12 weeks. The fat content of HFD and LFD was 60% and 10%, respectively. From the 13th week, oleic acid or palmitoleic acid conjugated with gavage bovine serum albumin (BSA) was gavaged into mice at a dose of 600 mg/d/kg body weight. BSA alone was used as the control. The duration of the intervention was 6 weeks. The experiment was divided into 6 groups (10 mice per group): low-fat control group (LFD BSA), low-fat oleic acid group (LFD oleic acid), low-fat palmitoleic acid group (LFD palmitoleic acid), high-fat control group (HFD BSA), high-fat oleic acid group (HFD oleic acid), and high-fat palmitoleic acid group (HFD palmitoleic acid). Body weight and food consumption were recorded at a fixed time every week. Glucose tolerance tests (GTT), insulin tolerance tests (ITT), and pyruvate tolerance (PTT) tests were performed at the 15th, 16th, and 17th weeks, respectively. A total of 40 C57BL/6N male mice aged 5–6 weeks were injected with AAV-SIRT3 virus or AAV-GFP virus as a control through a tail vein. Mice were then fed with HFD or LFD for a total of 13 weeks. There were 4 experimental groups: low-fat diet control group (LFD AAV-GFP), low-fat diet SIRT3 overexpression group (LFD AAV-SIRT3), high-fat diet control group (HFD AAV-GFP), high-fat diet SIRT3 overexpression group (HFD AAV-SIRT3). Body weight and food consumption were recorded at a fixed time every week. GTT and PTT were performed at 12th and 13th weeks, respectively. When mice were euthanized, serum and tissue samples were collected. After weighing, liver, adipose tissue (epididymal fat and inguinal fat), and muscle (gastrocnemius muscle, tibialis anterior muscle, and extensor digitorum longus muscle) were frozen in liquid nitrogen and then stored at -80 °C for further analyses [23]. All procedures were approved by the Institutional Animal Care and Use Committee at Shandong University and performed in conformance with the guide.

### 2.3. GTT, ITT, and PTT

GTT, ITT, and PTT were performed as previously described [24,25]. Before the experiment, mice were fasted overnight (for GTT and PTT) or for 4 h (for ITT). Glucose (2 g/kg), insulin (1.0 U/kg), or pyruvate (2 g/kg) was injected intraperitoneally. Blood glucose concentration from tail vein was measured at 0 (before injection), 30, 60, 90, and 120 min; 15, 30, 60, and 120 min (for GTT or PTT); or 15, 30, 45, and 60 min (for ITT) using a glucose meter.

### 2.4. HOMA-IR Calculation

Blood from fasted mice was collected and centrifuged at 12,000 rpm at 4 °C for 5 min. Sera were collected to measure fasting blood glucose and insulin. Fasting blood glucose was measured using a glucose assay kit (glucose oxidase method) (Jingmei, #F006-1-1). Insulin level was measured using a mouse insulin ELISA kit (Invitrogen, #EMINS 96 test). The formula of HOMA-IR was the level of fasting blood glucose (mmol/L) multiplied by the level of fasting insulin (microU/mL) divided by 22.5.

### 2.5. Western Blot Analysis

Lysates were prepared from frozen tissue samples or cultured cells. Western blots were conducted. The levels of SIRT3 (Cell Signaling Technology, Danvers, MA, USA, #5490), SIRT1 (Cell Signaling Technology, Danvers, MA, USA, #9475S), PEPCK (PCK1, Cell Signaling Technology, Danvers, MA, USA, #12940), PC (Novus, Littleton, CO, USA, #49536), MDH2 (Cell Signaling Technology, Danvers, MA, USA, #8610), PHB2 (Cell Signaling Technology, Danvers, MA, USA, #14085), GAPDH (Cell Signaling Technology, Danvers, MA, USA, #5174), anti-FLAG (Cell Signaling Technology, Danvers, MA, USA, #14793), and Tubulin (Cell Signaling Technology, Danvers, MA, USA, #2125) were detected as previously described [26].

### 2.6. Real-Time Quantitative Polymerase Chain Reaction (RT-PCR)

Total RNA was extracted from the liver with TRIzol and purified by miRNeasy Micro Kit (Qiagen, Germantown, MD, USA, #217084). cDNA was obtained by reverse transcription from total RNA using SuperScript IV Reverse Transcriptase (Invitrogen, Waltham, MA, USA, #18090010), and mRNA expression levels were analyzed by Roche LC480 fluorescence quantitative PCR (#28424) using Platinum™ Taq DNA Polymerase (Invitrogen, #15966005). To detect the expression of SIRT3 in the liver, the following primers were used. The forward primer is CGGCTCTATACACAGAACATCGA and the reverse primer is GTGGGCTTCAACCAGCTTTG.

### 2.7. Enzyme Activity Determination

PEPCK and PC enzyme activities were measured using the following kits from Solarbio, Beijing, China: PEPCK (#BC3315) and PC (#BC0735). Protein was extracted from liver tissue without protease inhibitors. The NADH decline rate was measured at 340 nm. When the reaction time was 10 s, the absorbance value was A1. When the reaction time was 1 min and 30 s, the absorbance value was A2. The calculation formula is as follows:ΔA = (A1 measured value–A2 measured value) − (A1 blank–A2 blank)PEPCK enzyme activity (U/mg prot) = 3215.4 × ΔA÷CprPC enzyme activity (U/mg prot) = 1607 × ΔA ÷ CprCpr is sample protein concentration (mg/mL).

A MDH2 activity assay kit (Abcam, Cambridge, UK, #ab119693) was used to measure MDH2 enzymatic activity. Protein was extracted from liver tissue without protease inhibitors. The activity was expressed as the change in absorbance per minute per amount of sample (∆ mOD/min/mg). Total protein concentration was measured using a BCA protein assay kit.

### 2.8. Primary Hepatocytes Isolation

All buffers and media were prepared fresh. Next, 1 × perfusion buffer 1 and 1 × perfusion buffer 2 were pre-warmed at 37 °C. The M199 medium mixture was ice-cold. A perfusion pump (Longer Pump LSP01-1A) and a 50 mL syringe with 1 × perfusion buffer 1 was set up. Mice were anesthetized using a ketamine and xylazine cocktail (contained ketamine 100 mg/mL, xylazine 20 mg/mL, and buprenorphine 0.3 mg/mL in PBS or saline). Mice were immobilized with tape, and the abdominal cavity was opened. A catheter (0.55 mm × 19 mm) was inserted into the portal vein. Mice were perfused with 1 × perfusion buffer 1 diluted from 10 × perfusion buffer 1 stock (pH 7.4, contained 1.42 M NaCl, 0.067 M KCl, 0.1 M HEPES, stored in 4 °C) with EGTA 2.5 mM, pH 7.4. As the buffer flowed and reached the liver, the inferior vena cava was cut. Perfusion buffer 1 flew at a rate of 5 mL/min for 5 min. This was followed with perfusion of perfusion buffer 2 containing 66.7 mM NaCl, 6.7 mM KCl, 100 mM HEPES, 6 mM CaCl2 2H_2_O, pH 7.6 (before use, 0.5 mg/mL collagenase (Sigma, St. Louis, MO, USA, #C5138) and 10 mg/mL fatty-acid-free albumin (Sigma, #B2064) were added) for 7–8 min. The liver was then removed and put into a dish containing perfusion buffer 2. Hepatocytes were released by shaking the liver using forceps. A volume of 10 mL M199 medium mixture (containing Medium 199 (Gibco, Lafayette, CO, USA, #31100-027), 23 mM HEPES, 26 mM Na bicarbonate, 100 units/mL penicillin, 100 μg/mL streptomycin, 10% FBS, 100 nM dexamethasone, 100 nM insulin, and 11 mM D-Glucose, pH7.4) was added into the dish. Liver cell suspension was passed through a strainer (Falcon 70 μm, Corning, NY, USA). Hepatocytes were centrifuged at 50 g for 4 min at 4 °C. The pellet was washed once with M199 medium mixture and resuspended in M199 medium mixture. Liver cell suspension was added into Percoll solution (90% Percoll (Sigma, St. Louis, MO, USA, #P1644), 9% 10 × PBS, 1% 1 M HEPES pH7.4) at a 1:1 ratio by carefully layering liver suspension onto a Percoll cushion. Tubes were centrifuged at 270 g for 5 min at 4 °C. Viable hepatocytes pelleted at the bottom of the tubes were washed once with M199 medium mixture. Resuspended hepatocytes were counted and plated into coated plates. The cells were used the next day [8].

### 2.9. Glucose Output Measurement

Primary mouse hepatocytes were freshly plated in 6-well plates in the maintenance medium (35 mm dish size, 4 × 10^5^ cells). Before the experiment started, the cells were washed twice with warm PBS and rinsed with no glucose and no phenol red DMEM (Thermo Fisher Scientific, Lafayette, CO, USA, Cat#A1443001). Then, the medium was replaced with 1 mL glucose-free DMEM without phenol red supplemented with 20 mM sodium lactate + 2 mM sodium pyruvate. After the cells were incubated for 3 h with or without 100 nM glucagon (Sigma, St. Louis, MO, USA, Cat# G2044), 0.1 mL of medium was collected, and the glucose concentration was measured using a hexokinase-based glucose assay. Glucose standards and samples were incubated with assay buffer mix (150 mM HEPES, 2.5 mM NADP, 2.5 mM ATP, 2.5 U/mL G6PDH, 5 U/mL hexokinase) for 10 min, and the absorbance at wavelength 340 nm was measured. Glucose concentration in the samples was calculated based on the standards (0~800 μM) and then normalized to the total protein content measured by a BCA protein assay kit.

### 2.10. Statistical Analysis

Data are presented as mean ± standard error of the mean (SEM). One-way ANOVA and least significance difference (LSD) method as a suitable post hoc test were used to determine the differences among groups using SPSS 26 (IBM, SPSS, Armonk, NY, USA), and a *p*-value of < 0.05 was considered statistically significant.

## 3. Results

### 3.1. Palmitoleic Acid Increases Systemic Glucose Clearance, Reduces Gluconeogenesis and the Expression of SIRT3 in the Liver of Obese Mice

Our previous study found that HFD-fed mice treated with palmitoleic acid had reduced HFD-induced liver insulin resistance and inflammation [8]. In this study, mice were fed with LFD or HFD for 12 weeks, then gavaged with BSA, oleic acid, or palmitoleic acid for 6 weeks. HFD significantly increased body weight, liver weight, and fat content in mice. Neither oleic acid nor palmitoleic acid significantly changed body weight, liver weight, or fat content in either LFD-fed mice or HFD-fed mice (data not shown). In LFD-fed mice, GTT revealed that palmitoleic acid significantly increased systemic glucose clearance compared to the BSA control or oleic acid treatment (Figure 1A,B). However, ITT (Figure 1E,F), HOMA-IR (Figure 1M), and PTT (Figure 1I,J) did not show differences among LFD groups. In HFD-fed mice, palmitoleic acid treatment resulted in statistically significant increases in systemic glucose clearance (Figure 1C,D) and an increase in systemic insulin sensitivity (Figure 1G,H), as well as a significant decrease in glucose production compared with the BSA and oleic acid groups (Figure 1K,L), indicating that reduction in hepatic gluconeogenesis might play a role in enhanced systemic glucose homeostasis by palmitoleic acid in obese mice.

Many studies have found that palmitoleic acid increases insulin sensitivity [27,28]. We found that palmitoleic acid also significantly increased systemic glucose clearance by reducing hepatic gluconeogenesis. Next, we investigated how palmitoleic acid reduces hepatic gluconeogenesis. SIRT3 is a mitochondrial enzyme that plays an important role in energy homeostasis [29,30]. SIRT3 expression is increased in the liver during nutrient deprivation, such as under fasting or caloric restriction [31,32]. As SIRT3 regulates many metabolic responses to fasting, such as ketogenesis [33], SIRT3 might also regulate gluconeogenesis, which is activated by fasting. Therefore, we detected the expression of SIRT3 in the liver of mice treated with BSA, oleic acid, or palmitoleic acid. There was no difference in the liver expression of SIRT3 among mice fed with LFD (Figure 2A,B), whereas the expression of SIRT3 in the liver of mice treated with palmitoleic acid was significantly reduced under HFD feeding (Figure 3A,B). SIRT1 was reported to be involved in enhancing gluconeogenesis by increasing the transcription of the key gluconeogenic genes, such as PCK1(PEPCK), via deacetylation of signal transducer and activator of transcription 3 (STAT3) [34]. Therefore, we also detected the expression of SIRT1 and found that palmitoleic acid did not change SIRT1 expression in obese mice (Figure 3A,F).

### 3.2. Palmitoleic Acid Does Not Change the Liver Protein Levels of PC and PEPCK, but Reduces the Level of MDH2 and the Enzymatic Activities of PC, PEPCK, and MDH2 in Obese Mice

During gluconeogenesis, in mitochondria, pyruvate converts to oxaloacetate by pyruvate carboxylase (PC), whereas malate dehydrogenase 2 (MDH2) catalyzes oxaloacetate to malate. Malate then enters cytosol and converts to oxaloacetate, which is the substrate of phosphoenolpyruvate carboxykinase (PEPCK), to produce phosphoenolpyruvate [35,36]. Here, we found that in the liver, palmitoleic acid did not alter the expressions of PC and PEPCK. But it decreased the expression of MDH2 (Figure 3A,C–E). Many studies have shown SIRT3-mediated deacetylation can regulate the activities of many enzymes [37]. Therefore, we speculated that the deacetylase property of SIRT3 might play a role in the process of gluconeogenesis by altering the activities of the gluconeogenic enzymes. Indeed, under HFD, palmitoleic acid significantly reduced the activities of MDH2, PC, and PEPCK (Figure 4A–C). The results showed that although during HFD feeding, palmitoleic acid did not change the protein levels of PC and PEPCK, the activities of PC and PEPCK were decreased.

### 3.3. Liver-Specific SIRT3 Overexpression Increases Hepatic Gluconeogenesis in Mice

To investigate whether SIRT3 regulates hepatic gluconeogenesis, we generated mice with liver-specific overexpression of SIRT3. Mice were injected with AAV-GFP or AAV-SIRT3 (with FLAG tag) through a tail vein. Mice were fed with LFD or HFD for 12 weeks. SIRT3 was overexpressed in the livers of AAV-SIRT3 mice under LFD or HFD at both the protein level (Figure 5A) and the mRNA level (Figure 5B). It is worth noting that the high-fat diet did not alter SIRT3 expression in the liver. Exogenous SIRT3 overexpression was not observed in adipose tissues or muscle (Figure 5C–F). However, HFD had a trend of reducing endogenous SIRT3 expression in adipose tissue, especially in the AAV-SIRT3 group (Figure 5C,D). PTT showed that HFD feeding significantly increased hepatic gluconeogenesis, with increased blood glucose levels 15, 30, and 60 min after pyruvate injection compared with LFD feeding (Figure 5G,H). SIRT3 overexpression increased hepatic gluconeogenesis under both HFD and LFD (Figure 5G,H). In addition, HFD significantly increased body weight, fat content, and liver weight in mice (Appendix A), whereas SIRT3 overexpression did not change body weight, fat content, or liver weight in mice under either LFD or HFD (Appendix A). Neither HFD nor SIRT3 overexpression affected the weight of the tibialis anterior muscle (TA), extensor digitorum longus muscle (EDL), gastrocnemius muscle (Gastroentero), or soleus muscle (Appendix A).

### 3.4. Liver-Specific SIRT3 Overexpression Does Not Alter the Protein Levels of PC and PEPCK, but Increases MDH2 Protein Levels in Mice

Examination of the expression of gluconeogenesis-related enzymes, such as PEPCK, PC, and MDH2 in the liver revealed that HFD significantly increased the expression of MDH2 (Figure 6A,D), but not the expression of PEPCK and PC (Figure 6A–C). Under both LFD and HFD feeding, SIRT3 overexpression did not change the expression of PEPCK and PC (Figure 6A–C). Under LFD feeding, SIRT3 increased the expression of MDH2, whereas under HFD feeding, SIRT3 also had a tendency to increase the expression of MHD2, although it was not statiscally significant (Figure 6D). These results showed that SIRT3 overexpression did not change the protein levels of PEPCK and PC, but increased the protein level of MDH2.

### 3.5. Liver-Specific SIRT3 Overexpression Increases the Activities of Gluconeogenic Enzymes

The above results indicate that palmitoleic acid decreased gluconeogensis and suppressed the expression of SIRT3 and MDH2 in obese mice. Moreover, SIRT3 overexpression increased gluconeogenesis and the expression of MDH2 but not PEPCK and PC. We then investigated whether SIRT3 regulates the activities of gluconeogenic enzymes. In the livers of AAV-infected mice fed with LFD or HFD, we found that HFD significanlty increased the enzymatic activity of MDH2 but not the enzymatic activity of PEPCK or PC (Figure 7A–C). SIRT3 overexpression significantly increased the enzymatic activities of PEPCK and PC under both LFD and HFD, as well as increasing the enzymatic activity of MDH2 under LFD (Figure 7A–C). These results suggest that SIRT3 overexpression can increase hepatic gluconeogenesis by increasing the enzyme activities of gluconeogenic enzymes MDH2, PEPCK, and PC.

### 3.6. SIRT3 Oerexpression Increases Glucose Production and the Protein Level of MDH2 in Primary Hepatocytes

Adenoviral vectors Ad-GFP and Ad-SIRT3 (with the GFP) were used to infect primary hepatocytes. Primary hepatocytes from mice fed with HFD or LFD for 12 weeks were isolated and infected with adenoviruses. After infection, the efficiency was observed under a fluorescence microscope. After two days, the efficiency of adenovirus infection was significant (Figure 8A). A glucose production assay revealed that the glucose production of primary hepatocytes was reduced in the presence of the SIRT3 enzymatic activity inhibitor 3-TYP in primary hepatocytes from both LFD- and HFD-fed mice. SIRT3 overexpression increased glucose production in hepatocytes (Figure 8B). 3-TYP did not change the expression of SIRT3 but inhibited the enzyme activity of SIRT3. Neither SIRT3 inhibitor nor SIRT3 overexpression changed the protein levels of PEPCK (data not shown) or PC (Figure 8C). SIRT3 overexpression increased the protein level of MDH2 (Figure 8C).

### 3.7. Palmitoleic Acid Reverses the Increasing of Gluconeogenesis Mediated by SIRT3 Overexpression in Primary Hepatocytes

Primary hepatocytes from mice fed with HFD for 12 weeks were isolated and infected with Ad-GFP or Ad-SIRT3 (with the GFP). Cells were treated with palmitoleic acid for 48 h. In the presence of substrates and glucagon, SIRT3 overexpression increased glucose production in primary hepatocytes, and palmitoleic acid treatment blocked the increase in glucose production caused by SIRT3 overexpression (Figure 9A). Palmitoleic acid reduced SIRT3 expression in primary hepatocytes (Figure 9B). SIRT3 overexpression alone and combined with palmitoleic acid treatment did not significantly alter the protein levels of PC or PEPCK (Figure 9B). However, SIRT3 overexpression significantly increased the protein level of MDH2. Palmitoleic acid did not significantly reduce the increased protein level of MDH2 caused by SIRT3 overexpression (Figure 9B). These results indicate that palmitoleic acid abolished SIRT3-induced gluconeogensis by decreasing the protein level of SIRT3 without affecting the expression of the gluconeogenic enzymes.

## 4. Discussion

In summary, our study has demonstrated that palmitoleic acid reduced hepatic gluconeogenesis by decreasing SIRT3 expression and the enzymatic activities of gluconeogenic enzymes, such as PEPCK, PC, and MDH2.

As we know, saturated fat has a greater stimulating effect on hepatic lipid accumulation and inflammation than unsaturated fat [38,39]. Nutritional stress, especially excessive dietary fats, leads to obesity and insulin resistance. This process is believed to be the result of impaired insulin signal transduction caused by several molecules released from adipose tissue [40,41]. However, not all fatty acids have a negative effect on insulin sensitivity [42]. Several studies found that palmitoleic acid has a beneficial effect of increasing insulin sensitivity [43,44]. It has also been found that palmitoleic acid enhances the liver and systemic insulin sensitivity in humans [45,46,47]. In this study, we found that under LFD, palmitoleic acid increased systemic glucose clearance compared to BSA control or oleic acid treatment. However, it did not enhance insulin sensitivity or inhibit gluconeogenesis. We think there are two possible reasons for this phenotype. First, when we inject glucose intraperitoneally, glucose is absorbed from the peritoneal serous membranes, enters the blood stream, then follows the portal system to the liver. In LFD-fed mice treated with palmitoleic acid, glucose may be trapped in the liver to synthesize glycogen; therefore, less glucose is released into circulation. Secondly, because palmitoleic acid can regulate pancreatic β-cell insulin secretion [42], when glucose is intraperitoneally injected into LFD-fed mice with palmitoleic acid treatment, pancreatic β cells may release more insulin to enhance systemic glucose clearance [48]. Palmitoleic acid supplementation also significantly increased glucose clearance in HFD-induced obese mice. In addition, we found that palmitoleic acid significantly reduced hepatic gluconeogenesis in HFD-fed mice. Therefore, the downregulation of hepatic gluconeogenesis may contribute to the systemic glucose homeostasis under the HFD condition. In this study, we found that palmitoleic acid only modestly improves glucose homeostasis under the low-fat condition, whereas its effects are more pronounced in HFD-fed mice. This is likely due to the fact that under the low-fat diet condition, the mice were insulin sensitive. There is not much room to further improve glucose homeostasis by palmitoleic acid. Under an HFD diet, palmitoleic acid significantly inhibited gluconeogenesis and improved glucose homeostasis. Furthermore, palmitoleic acid reduced the expression of SIRT3 and MDH2 in the liver, without changing the expression of key gluconeogenic enzymes PEPCK and PC. However, palmitoleic acid suppresses the enzymatic activities of PC, PEPCK, and MDH2. Therefore, it is likely that palmitoleic acid inhibits gluconeogenesis through the suppression of SIRT3 expression and the activities of gluconeogenic enzymes.

To study the effect of SIRT3 overexpression on gluconeogenesis, we overexpressed SIRT3 specifically in the liver of mice. Under HFD and LFD conditions, the specific overexpression of SIRT3 in the liver increases gluconeogenesis. In vitro, we also found that overexpression of SIRT3 enhances the glucose production by primary hepatocytes. Inhibiting the deacetylase activity of SIRT3 reduces glucose production. Our data show that HFD can increase the expression of MDH2 but not PEPCK and PC. However, other studies have found [49] that HFD does not change the expression of other enzymes in mitochondria, such as CS, PDHE1α, PDHK1, MT-ND1, SDHA1, and MnSOD. Enzyme acetylation modification plays an important role in regulating enzyme activities in glycolysis and gluconeogenesis [50]. We found that SIRT3 overexpression has a direct effect on the expression and activity of MDH2. MDH2 acts as a bridge connecting oxaloacetate into the gluconeogenesis process, crossing the mitochondrial membrane. SIRT3 regulates MDH2 activity [51]. Therefore, in addition to regulating the expression level of MDH2, SIRT3 may also regulate the enzymatic activities of MDH2 through deacetylation. In addition, overexpression of SIRT3 in the liver also increased the enzymatic activities of PC and PEPCK, without changing the protein levels of these two enzymes. SIRT3 appears to deacetylate PC. Proteomic studies have revealed that PC protein can be acetylated, and the level of PC acetylation is increased in SIRT3-deficient mice [32]. Under starvation conditions, the transcription of PEPCK is greatly increased. However, a 90% reduction in liver PEPCK protein levels only lead to a 40% reduction in gluconeogenesis [52], suggesting that in addition to the expression level of PEPCK, post-translational modification of PEPCK and changes of its enzyme activity might also contribute to the regulation of gluconeogenesis. In fact, PEPCK has been reported to be acetylated [53,54]. Therefore, SIRT3 may increase the enzymatic activities of PC and PEPCK by deacetylating them.

SIRT3 is mainly localized in mitochondria. We separated the cytoplasm and mitochondria of the liver to observe the difference in SIRT3 expression. Glyceraldehyde 3-phosphate dehydrogenase (GAPDH) is used as a marker for cytoplasm and prohibitin-2 (PHB2) for mitochondria. We observed some expressions of SIRT3 in cytoplasm under the LFD condition (Appendix A). Under the HFD condition, the levels of SIRT3 in the cytoplasm were increased (Appendix A). The key enzymes of gluconeogenesis, PC and PEPCK, are located in the mitochondria and the cytoplasm, respectively. Therefore, SIRT3 might regulate PC and PEPCK activities in mitochondria and cytosol, especially under the HFD condition.

## 5. Conclusions

Overall, in this study, we found that SIRT3 plays a role in post-translational modification of gluconeogenic enzymes and increases their enzymatic activities. Targeting hepatic SIRT3 to reduce gluconeogenesis may provide a novel treatment for diabetic. We also provided evidence indicating that palmitoleic acid decreases gluconeogenesis by reducing the enzymatic activities of gluconeogenic proteins by downregulating the expression of SIRT3.

## Figures and Tables

**Figure 1 nutrients-14-01482-f001:**
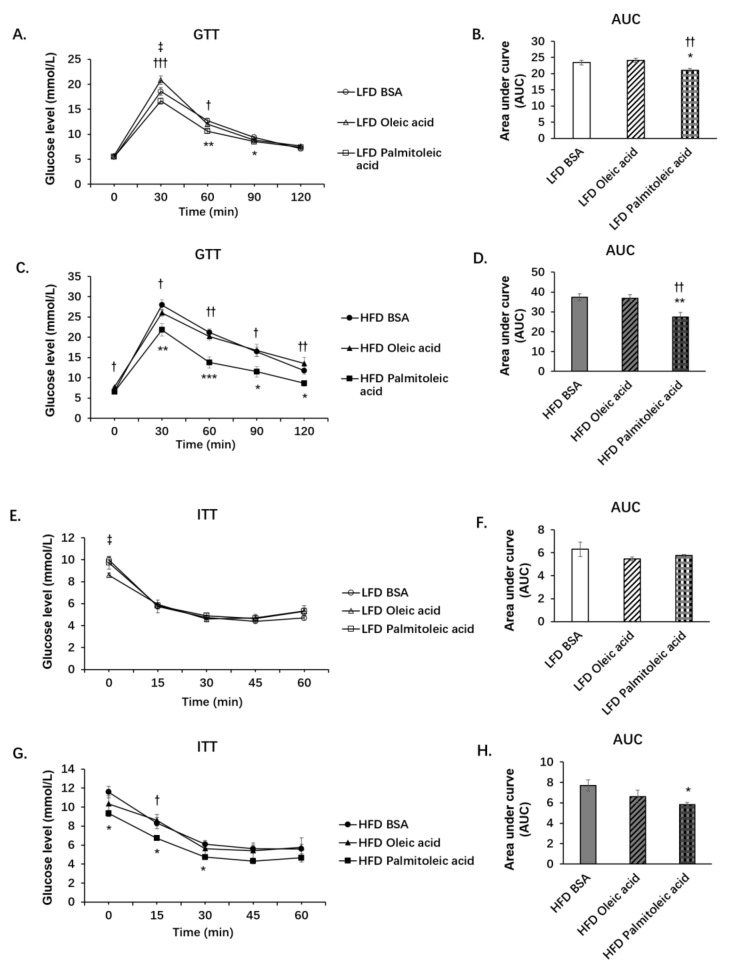
Palmitoleic acid increases systemic glucose clearance and reduces hepatic gluconeogenesis in obese mice. C57BL/6N male mice of 5–6 weeks of age were fed with LFD (low-fat diet) or HFD (high-fat diet) for 12 weeks. At the 13th week, BSA (bovine serum albumin), oleic acid, or palmitoleic acid was administered intragastrically. At the15th, 16th, and 17th weeks, GTT (Glucose tolerance tests), ITT (insulin tolerance tests) and PTT (pyruvate tolerance) were performed, respectively. (**A**) GTT for LFD group. (**B**) Area under curve of GTT in LFD group. (**C**) GTT for HFD group. (**D**) Area under curve of GTT in HFD group. (**E**) ITT for LFD group. (**F**) Area under curve of ITT in LFD group. (**G**) ITT for HFD group. (**H**) Area under curve of ITT in HFD group. (**I**) PTT for LFD group. (**J**) Area under curve of PTT in LFD group. (**K**) PTT for HFD group. (**L**) Area under curve of PTT in HFD group. Above data: *n* = 6–13 mice per group. (**M**) HOMA-IR in LFD group. (**N**) HOMA-IR in HFD group. Calculated based on fasting glucose level (mmol/L) and fasting insulin level (microU/mL). For HOMA-IR: *n* = 4–8 mice per group. The data are mean ± s.e. (error bars). *, *p* < 0.05, **, *p* < 0.01, ***, *p* < 0.001, BSA vs. palmitoleic acid; †, *p*< 0.05, ††, *p* < 0.01, †††, *p* < 0.001, oleic acid vs. palmitoleic acid; ‡, *p* < 0.05, ‡‡, *p* < 0.01, BSA vs. oleic acid.

**Figure 2 nutrients-14-01482-f002:**
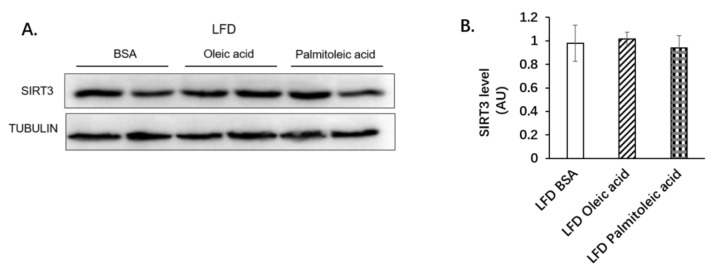
Palmitoleic acid did not alter SIRT3 expression in the liver of mice under LFD. C57BL/6N male mice of 5–6 weeks of age were fed with LFD for 12 weeks. At the 13th week, BSA, oleic acid, or palmitoleic acid was administered intragastrically. At the 18th week, mice were euthanized, and liver tissue was collected. (**A**) Expression of SIRT3 was detected using Western blot analysis. (**B**) Quantification of SIRT3 expression. AU, arbitrary units. The data are mean ± s.e. (error bars). *n* = 5 mice per group.

**Figure 3 nutrients-14-01482-f003:**
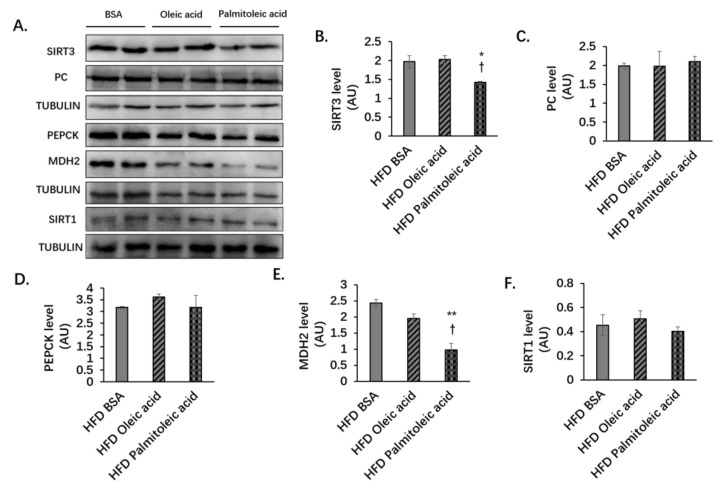
Palmitoleic acid reduced the expression of SIRT3 and MDH2 but did not change the protein levels of PC, PEPCK, and SIRT1 in the liver of mice under HFD. C57BL/6N male mice of 5–6 weeks of age were fed with HFD for 12 weeks. At the 13th week, BSA, oleic acid, or palmitoleic acid were administered intragastrically. At the 18th week, mice were euthanized, and liver tissues were collected. (**A**) Expression of SIRT3, SIRT1, and gluconeogenic enzymes in the liver of mice under HFD. Protein expression was detected using Western blot analysis. (**B**) Quantification of SIRT3 protein level. (**C**) Quantification of PC protein level. (**D**) Quantification of PEPCK protein level. (**E**) Quantification of MDH2 protein level. (**F**) Quantification of SIRT1 protein level. AU, arbitrary units. The data are mean ± s.e. (error bars). *n* = 4–6 mice per group. *, *p* < 0.05, **, *p* < 0.01, HFD BSA vs. HFD palmitoleic acid; †, *p* < 0.05, oleic acid vs. palmitoleic acid.

**Figure 4 nutrients-14-01482-f004:**
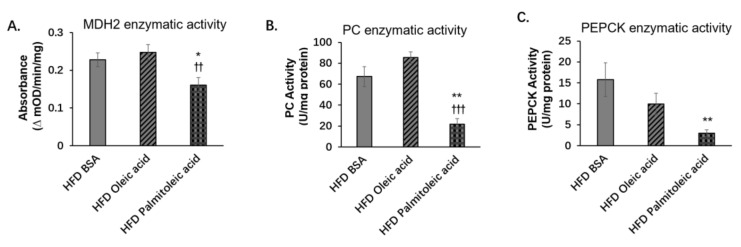
Palmitoleic acid reduces enzymatic activities of MDH2, PC, and PEPCK under HFD. C57BL/6N male mice of 5–6 weeks of age were fed with LFD or HFD for 12 weeks. At the 13th week, BSA, oleic acid, or palmitoleic acid was administered intragastrically. At the 18th week, mice were sacrificed, and liver tissue was collected. The enzyme activities of MDH2, PC, and PEPCK in the liver were measured. (**A**) Enzymatic activity of MDH2 in the liver of HFD-fed mice. (**B**) Enzymatic activity of PC in the liver of HFD-fed mice. (**C**) Enzymatic activity of PEPCK in the liver of HFD-fed mice. The data are mean ± s.e. (error bars). *n* = 4–7 mice per group. *, *p* < 0.05, **, *p* < 0.01, HFD BSA vs. HFD palmitoleic acid; ††, *p* < 0.01, †††, *p* < 0.001, HFD oleic acid vs. HFD palmitoleic acid.

**Figure 5 nutrients-14-01482-f005:**
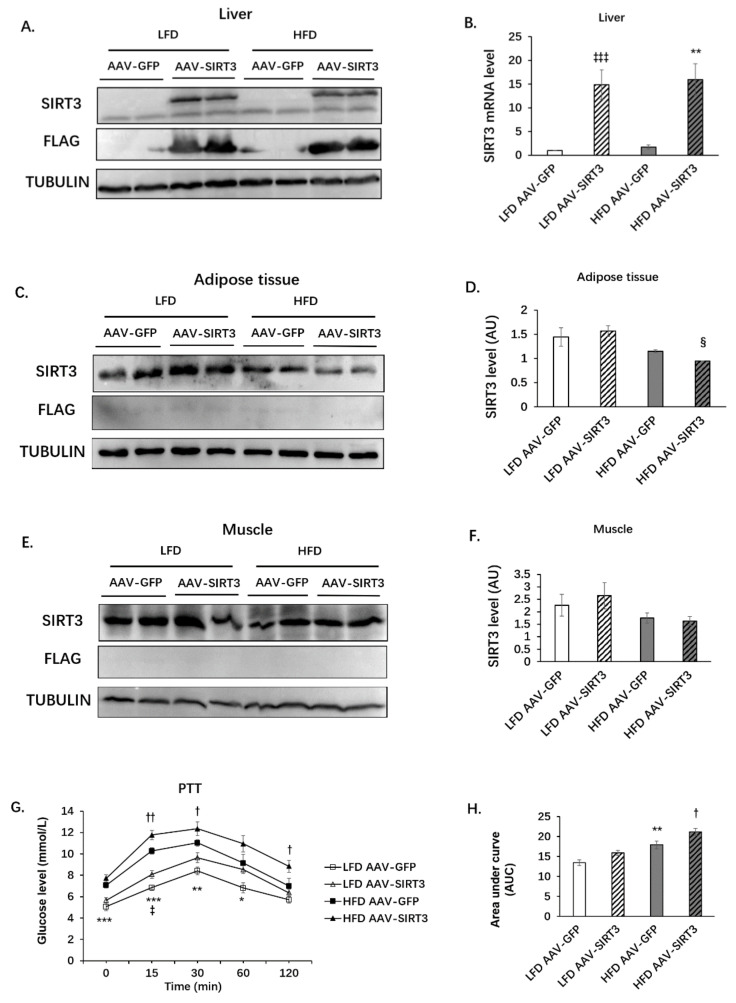
Liver-specific SIRT3 overexpression increases gluconeogenesis. C57BL/6N male mice of 5–6 weeks of age were injected with AAV-GFP or AAV-SIRT3 via a tail vein and fed with LFD or HFD for 12 weeks, respectively. At the 13th week, PTT was performed. At the 14th week of feeding, mice were euthanized, and liver tissues were collected. (**A**) Protein level of SIRT3 in liver. (**B**) RNA level of SIRT3 in liver. (**C**) Protein level of SIRT3 in adipose tissue. (**D**) Quantification of SIRT3 protein level in adipose tissue. (**E**) Protein level of SIRT3 in muscle. (**F**) Quantification of SIRT3 protein level in muscle. The data are mean ± s.e. (error bars). *n* = 6 mice per group. **, *p* < 0.01, AAV-GFP vs. AAV-SIRT3. (**G**) PTT. Mice were fasted overnight and intraperitoneally injected with pyruvate (2 g/kg). Blood glucose levels were measured before injection and 15, 30, 60, and 120 min after injection. (**H**) Area under curve of PTT. The data are mean ± s.e. (error bars). *n* = 6–9 mice per group. §, *p* < 0.05, LFD AAV-SIRT3 vs. HFD AAV-SIRT3; *, *p* < 0.05, **, *p* < 0.01, ***, *p* < 0.001, LFD AAV-GFP vs. HFD AAV-GFP; ‡, *p* < 0.05, ‡‡‡, *p* < 0.001, LFD AAV-GFP vs. LFD AAV-SIRT3; †, *p* < 0.05, ††, *p* < 0.01, HFD AAV-GFP vs. HFD AAV-SIRT3.

**Figure 6 nutrients-14-01482-f006:**
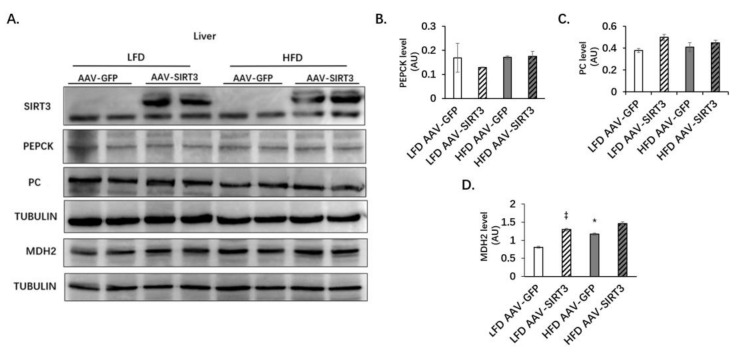
Liver-specific SIRT3 overexpression increases MDH2 protein levels in mice. C57BL/6N male mice of 5–6 weeks of age were injected with AAV-GFP or AAV-SIRT3 via a tail vein and fed with LFD or HFD for 12 weeks. At the 14th week of feeding, mice were euthanized, and the liver tissues were collected. (**A**) Expression of SIRT3 and the levels of gluconeogenic-related enzymes in the liver. (**B**) Quantitative the expression of PEPCK protein level. (**C**) Quantitative the expression of PC protein level. (**D**) Quantification the expression of MDH2 protein level. AU, arbitrary units. The data are mean ± s.e. (error bars). *n* = 6 mice per group. *, *p* < 0.05, LFD AAV-GFP vs. HFD AAV-GFP; ‡, *p* < 0.05, LFD AAV-GFP vs. LFD AAV-SIRT3.

**Figure 7 nutrients-14-01482-f007:**
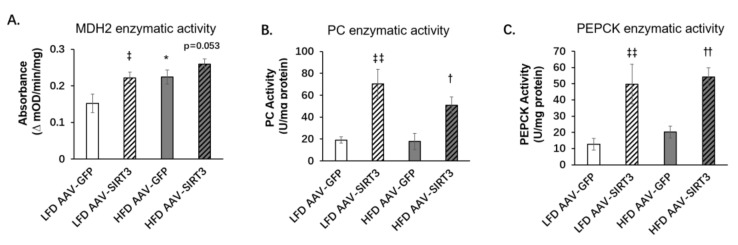
SIRT3 overexpression increases the activities of gluconeogenic enzymes. C57BL/6N male mice of 5–6 weeks of age were injected with AAV-GFP or AAV-SIRT3 via a tail vein and fed with LFD or HFD for 12 weeks. At the 14th week of feeding, mice were euthanized, and the liver tissues were collected. (**A**) Enzymatic activity of MDH2. (**B**) Enzymatic activity of PC. (**C**) Enzymatic activity of PEPCK. The data are mean ± s.e. (error bars). *n* = 4–7 mice per group *, *p* < 0.05, LFD AAV-GFP vs. HFD AAV-GFP; ‡, *p* < 0.05, ‡‡, *p* < 0.01, LFD AAV-GFP vs. LFD AAV-SIRT3; †, *p* < 0.05, ††, *p* < 0.01, HFD AAV-GFP vs. HFD AAV-SIRT3.

**Figure 8 nutrients-14-01482-f008:**
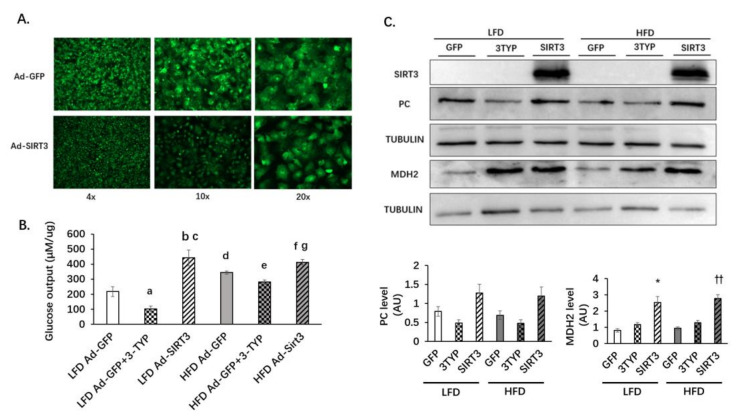
SIRT3 overexpression increases glucose production and the protein level of MDH2 in primary hepatocytes. Primary hepatocytes from HFD or LFD mice were isolated and infected with Ad-GFP or Ad-SIRT3 adenovirus. (**A**) Infection status after 48 h as fluorescent detection of GFP. (**B**) Glucose production from primary hepatocytes. In the presence of substrates (20 mM lactic acid and 2 mM pyruvate), the primary hepatocytes were stimulated with 100 nM glucagon for 3 h with or without 50 uM 3-TYP treatment for 24 h. a, *p* < 0.05, LFD Ad-GFP vs. LFD Ad-GFP+3-TYP; b, *p* < 0.05, LFD Ad-GFP vs. LFD Ad-SIRT3; c, *p* < 0.05, LFD Ad-GFP+3-TYP vs. LFD Ad-SIRT3; d, *p* <0.05, LFD Ad-GFP vs. HFD Ad-GFP; e, *p* < 0.05, HFD Ad-GFP vs. HFD Ad-GFP+3-TYP; f, *p* < 0.05, HFD Ad-GFP vs. HFD Ad-SIRT3; g, *p* < 0.01, HFD Ad-GFP+3-TYP vs. HFD Ad-SIRT3. (**C**) SIRT3, PC and MDH2 protein levels. Primary hepatocytes infected with adenovirus were treated with DMSO or 3-TYP for 24 h and 100 nM glucagon for 1 h. Cells were collected for protein extraction. Western blot analysis was used to detect the expression of SIRT3, PC, and MDH2. *, *p* < 0.05, LFD Ad-GFP vs. LFD Ad-SIRT3; ††, *p* < 0.01 HFD Ad-GFP vs. HFD Ad-SIRT3. The data are mean ± s.e. (error bars). *n* = 5 from three independent experiments.

**Figure 9 nutrients-14-01482-f009:**
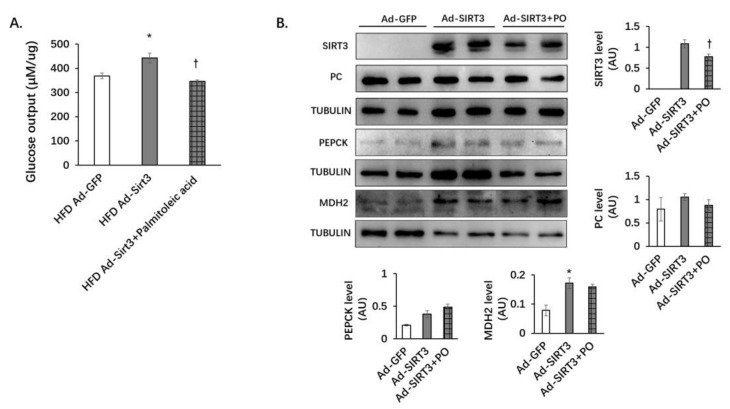
Palmitoleic acid reverses the increase in gluconeogenesis caused by SIRT3 overexpression in primary hepatocytes. Primary hepatocytes from mice fed with HFD were isolated and infected with Ad-GFP or Ad-SIRT3 adenovirus. Ad-SIRT3-infected hepatocytes were also treated with palmitoleic acid for 48 h. (**A**) Glucose production from primary hepatocytes. In the presence of substrates (20 mM lactic acid and 2 mM pyruvate), the primary hepatocytes were stimulated with 100 nM glucagon for 3 h to detect the content of glucose in the medium. (**B**) SIRT3, PC, and MDH2 protein levels were detected using Western blot analysis. The data are mean ± s.e. (error bars). *n* = 6 from three independent experiments. *, *p* < 0.05, HFD Ad-GFP vs. HFD Ad-SIRT3; †, *p* < 0.05 HFD Ad-SIRT3 vs. HFD Ad-SIRT3+PO (palmitoleic acid).

## Data Availability

The datasets used and/or analyzed during the current study are available from the corresponding author on reasonable request.

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
