# Peer review of "The Role of Palmitoleic Acid in Regulating Hepatic Gluconeogenesis through SIRT3 in Obese Mice"

_nutrients, 2022, doi:10.3390/nu14071482_

Round 1
Reviewer 1 Report
Journal: Nutrients
Manuscript ID: nutrients-1628976
Type of manuscript: Article
Title: The role of palmitoleic acid in regulating hepatic gluconeogenesis through SIRT3 in obese mice
Authors: Xin Guo *, Xiaofan Jiang, Keyun Chen, Qijian Liang, Shixiu Zhang, Juan Zheng, Xiaomin Ma, Hongmei Jiang, Hao Wu, Chaodong Wu, Qiang Tong *
Summary: Palmitoleic acid treatment improved systemic glucose clearance in HFD fed C57BL/6N. It also reduced SIRT3 protein level and enzymatic activity of PEPCK, PC and MDH2 in liver of HFD fed C57BL/6N mice, suggesting that palmitoleic acid reduces SIRT3 to suppress gluconeogenic enzymatic activity. Hepatic SIRT3 overexpression enhanced gluconeogenic enzymatic activity and glucose production in primary hepatocytes. Although it is interesting study, further clarification is necessary to support the conclusion.
Comments:
- HFD-palmitoleic acid showed improved glucose tolerance (Fig. 1C). However, HFD-palmitoleic acid did not show improved insulin sensitivity (Fig. 1G) as the difference of AUC (Fig. 1F) was caused by the low fasting glucose of HFD-palmitoleic acid. If Fig. 1G was plotted with percentage, HFD-BSA and HFD-palmitoleic acid could not show any difference, suggesting that there is no difference in insulin sensitivity. 1U/Kg for ITT should be too high concentration to show the difference. Lower insulin such as 0.75U/Kg can show some difference.
Improved insulin sensitivity shows increased AKT phosphorylation. Measure AKT phosphorylation for Thr308 and Ser473 in liver and muscle after insulin injection.
- SIRT1 is also involved in glucose metabolism through deacetylation of gluconeogenic enzymes such as PCK1. HFD-palmitoleic acid showed reduced SIRT3 in Western blot (Fig. 3A). What is the SIRT1 protein level?
- “Here, we found that in the liver palmitoleic acid did not reduce the expressions of glycogenic pathway related enzymes, such as PC, PEPCK, and MDH2 (Fig. 3A, C, D, and E)” in lines 305-307. This description is not correct. PEPCK and MDH2 proteins, especially MDH2 (Fig. 3E), were decreased in HFD-palmitoleic acid.
- SIRT3 expression was decreased with HFD (Fig. 5C, D, E and F) although authors described that there is no difference in SIRT3 levels (lines 331-332). Show statistics especially for quantification (Fig. 5D and F).
- PEPCK protein was significantly increased with HFD in Fig. 6B. However, PECPCK activity did not show difference between LFD AAV-Sirt3 and HFD AAV-sirt3 in Fig. 7C. How can you explain this? Add explanation in discussion. It has been shown that acetylation is one of critical mechanisms to regulate PCK1 enzymatic activity. Is pyruvate carboxylase also regulated by acetylation? If not, how does SIRT3 overexpression increased PC activity in Fig. 7B?
- Isolation of primary hepatocytes from HFD fed mice is quite challenging as most fat accumulated primary hepatocytes from HFD fed mice were destroyed in isolation. Thus, it is not clear if results in Fig. 8 and 9 were reliable to present HFD effects in cell culture. Authors can replace results with primary hepatocytes prepared from LFD fed mice for Fig. 9.
- Change “glycogenic pathway” (line 306) into “gluconeogenic pathway”.
- Add tick marks for figure graphs.
- Describe fat% of LFD and HFD in methods.
- Spelling error in AAG-TBG-eGFP (line 107).
- Remove “and HFD” in line 284 and “LFD and” in line 292.
Reviewer 2 Report
The authors have elucidated mechanism of how palmitoleate reduces blood glucose and whether SIRT3 also regulates gluconeogenesis. The study revealed that palmitoleic acid reduced hepatic gluconeogenesis and the expression of SIRT3 under high-fat diet condition. The authors have further suggested that SIRT3 plays a role in enhancing enzymatic activities of gluconeogenic enzymes such as PEPCK, PC, and MDH2 concluding the study as under high fat diet, palmitoleic acid decreased gluconeogenesis by reducing enzymatic activities of PEPCK, PC, and MDH2 via down-regulating the expression of SIRT3. The manuscript is well written and discussed.
Minor concerns:
- In Fig 1N, the P-value of 0.054 is not statistically significant. The authors should change all the content related to this result.
- The authors need to proof-read the manuscript for several grammatical errors.
Author Response
We thank the reviewers for their constructive review of our manuscript. We have addressed their concerns point-by-point below and modified our manuscript accordingly. We marked up the changes in the revised manuscript. We hope that we have satisfactorily addressed the concerns of the reviewers and the manuscript is now suitable for publication in Nutrients.
Reviewer 2:
Comments and Suggestions for Authors
The authors have elucidated mechanism of how palmitoleate reduces blood glucose and whether SIRT3 also regulates gluconeogenesis. The study revealed that palmitoleic acid reduced hepatic gluconeogenesis and the expression of SIRT3 under high-fat diet condition. The authors have further suggested that SIRT3 plays a role in enhancing enzymatic activities of gluconeogenic enzymes such as PEPCK, PC, and MDH2 concluding the study as under high fat diet, palmitoleic acid decreased gluconeogenesis by reducing enzymatic activities of PEPCK, PC, and MDH2 via down-regulating the expression of SIRT3. The manuscript is well written and discussed.
Response: Thank you very much for your positive comments on our manuscript.
Minor concerns:
In Fig 1N, the P-value of 0.054 is not statistically significant. The authors should change all the content related to this result.
The authors need to proof-read the manuscript for several grammatical errors.
Response: Thank you for your comments. We corrected the content of the result. Please check result 3.1 in line 247-255, line 277-280, and in discussion line 502-505. We corrected several grammatical errors as well.
